# Determination of Hemicellulose, Cellulose, and Lignin Content in Different Types of Biomasses by Thermogravimetric Analysis and Pseudocomponent Kinetic Model (TGA-PKM Method)

**David Díez [1,2,*], Ana Urueña [1,2], Raúl Piñero [1,2], Aitor Barrio [3] and Tarja Tamminen [4]**

[1] CARTIF Centre of Technology, Parque Tecnológico de Boecillo, 205, Boecillo, 47151 Valladolid, Spain; anauru@cartif.es (A.U.); raupin@cartif.es (R.P.)

[2] ITAP Institute, University of Valladolid, Paseodel Cauce 59, 47011 Valladolid, Spain

[3] TECNALIA, Basque Research and Technology Alliance (BRTA), Área Anardi 5, E-20730 Azpeitia, Spain; aitor.barrio@tecnalia.com

[4] VTT-Technical Research Centre of Finland, P.O. Box 1000, VTT, FI-02044 Espoo, Finland; tarja.tamminen@vtt.fi

\* Correspondence: davdie@cartif.es

**Abstract:** The standard method for determining the biomass composition, in terms of main lignocellulosic fraction (hemicellulose, cellulose and lignin) contents, is by chemical method; however, it is a slow and expensive methodology, which requires complex techniques and the use of multiple chemical reagents. The main objective of this article is to provide a new efficient, low-cost and fast method for the determination of the main lignocellulosic fraction contents of different types of biomasses from agricultural by-products to softwoods and hardwoods. The method is based on applying deconvolution techniques on the derivative thermogravimetric (DTG) pyrolysis curves obtained by thermogravimetric analysis (TGA) through a kinetic approach based on a pseudocomponent kinetic model (PKM). As a result, the new method (TGA-PKM) provides additional information regarding the ease of carrying out their degradation in comparison with other biomasses. The results obtained show a good agreement between experimental data from analytical procedures and the TGA-PKM method (±7%). This indicates that the TGA-PKM method can be used to have a good estimation of the content of the main lignocellulosic fractions without the need to carry out complex extraction and purification chemical treatments. In addition, the good quality of the fit obtained between the model and experimental DTG curves ($R^2_{Adj} = 0.99$) allows to obtain the characteristic kinetic parameters of each fraction.

**Keywords:** TGA; hemicellulose; cellulose; lignin; pseudocomponent kinetic model; biomass

## 1. Introduction

The use of biomass resources for energy generation has been of considerable importance in recent years [1]. The global increase in energy demand has been one of the main reasons for their use. Added to this situation, there is also a need for dealing with certain problems, such as the depletion of fossil fuel reserves and the increase in environmental pollution from the use of these energy sources [2].

In this context, biomass has the advantage of being the only renewable resource that can be used in solid, liquid and gaseous forms [3]. Furthermore, biomass has the great capacity of producing by-products of high interest, such as catalytic carbons [4] and bioplastics [5]. However, biomass has

a number of features that make it difficult to use, including its moisture content, low-energy density and complex structure. Lignocellulosic biomass is made up of a structure that includes mainly cellulose, hemicellulose and lignin [6,7]. The proportions and distribution of these components in the biomass physical structure is complex and depends on the type of species. The knowledge of this composition is very important for its use in different industrial applications.

Up to now, the determination of biomass composition, in terms of hemicellulose, cellulose and lignin contents, has been made by the chemical method. However, it is a slow and expensive methodology, which requires complex techniques and the use of multiple chemical reagents [8]. This means that it is not a suitable method for use in industrial applications.

Thermogravimetric analysis and, especially, the derivative thermogravimetric (DTG) curve is often used for the preliminary study of various thermochemical processes with biomass, since it allows the determination of the different stages of biomass devolatilization. In general, the process of devolatilization of the biomass in the absence of oxygen usually differentiates four stages corresponding to the loss of moisture and the three lignocellulosic components (hemicellulose, cellulose and lignin) [3]. Numerous articles have been published in which the thermal decomposition intervals of these lignocellulosic components are presented based on the deconvolution of the DTG curves [9–15]. It has been observed that, after moisture removal that takes place up to 150 °C, the decomposition of the three biomass lignocellulosic components takes place: hemicellulose is the first component to decompose between 200–300 °C, followed by cellulose between 250–380 °C. Regarding the thermal decomposition of lignin, it is the component with the most complex structure, and its decomposition range is the widest [16], occurring from 200 °C up to high temperatures such as 1000 °C [17,18].

There are different studies based on determining the lignocellulosic composition by analyzing DTG curves. However, most of these studies are based exclusively on applying deconvolution methods without taking into account their kinetic interpretation of the process [3,19].

On the other hand, kinetic studies on the thermal decomposition of biomass are extensive, in which the use of different kinetic models is analyzed [20,21], providing the kinetic parameters that best fit the experimental data. However, these studies do not focus on finding a method that allows the quantification of the three main lignocellulosic fractions of the biomass.

The use of kinetic analysis to the quantification of the main lignocellulosic fractions allows to include restrictions for a more precise quantification, while a physical interpretation is added to the deconvolution process.

The main objective of this work is to provide a new efficient, low-cost and fast method for the determination of the hemicellulose, cellulose and lignin contents of different types of biomasses, from agricultural by-products to wood. The method is based on applying deconvolution techniques on DTG pyrolysis curves based on a kinetic analysis of the process, and the kinetic model used is based on the assumption that the degradation of each lignocellulosic fraction can be represented by the evolution of a certain number of pseudocomponents.

## 2. Materials and Methods

### 2.1. Biomass Samples

Five raw materials representing different types of biomass have been selected, including agricultural biomass (wheat straw) and forest biomass, both as softwood barks (spruce bark and pine bark) and hardwoods (poplar and willow).

The pine bark originated from Sweden, while the other biomasses (wheat straw, poplar, spruce bark and willow) came from the South of France.

## 2.2. Experimental Method

Each sample was crushed in a mill (Model A 10 basic, IKA-Werke GmbH & Co. KG, Staufen, Germany) and then sieved. The sample sizes were all less than 100 μm in order to minimize the heat transfer resistances and mass transfer diffusion effects.

The TG (thermogravimetric) analysis was performed on a TG-DTA analyzer (Model DTG-60H, SHIMADZU Co. Ltd., Kyoto, Japan). The analyses were carried out using a nitrogen atmosphere with a flow rate of 50 mL min$^{-1}$. The heating rate used was 5 °C min$^{-1}$, from room temperature to a final temperature of 1000 °C. The sample weight was c.a. 10 mg.

To reduce temperature-related errors, the equipment used was calibrated across the entire temperature range. In addition, the actual sample temperature was used directly to solve the kinetic equations and to calculate the actual sample heating rate [22].

The information obtained in these analyses was the weight loss as the temperature and time of analysis increase (TG curve).

## 2.3. Data Treatment

The TG analysis provides the weight loss as a function of temperature over time. The analysis can be used to determine the different fractions of volatiles released as a function of temperature, as well as the solid residue remaining after heat treatment. However, for the determination of kinetics, it is more useful to use the derivative thermogravimetric (DTG) of weight loss as a function of time, because this signal is much more sensitive to small changes.

Before proceeding with its calculation, it is necessary to preprocess the data in order to obtain a curve that depends exclusively on the process variables.

The first step is the normalization of the TG signal. The normalization has been carried out in relation to the initial weight of the sample ($m_0$) and the final weight ($m_\infty$) of the sample. To do this, the weight fraction of the volatiles remaining in the sample has been calculated for each instant of discrete time $i$, as indicated in Equation (1).

$$X_i = \frac{m_i - m_\infty}{m_0 - m_\infty} \tag{1}$$

In this case, $m_\infty$ represents the mass of char obtained at the end of each TG analysis and includes the mass of ash and fixed carbon at the final temperature of the analysis.

## 2.4. DTG Curves

The DTG curve is obtained from the weight over time derivative for each experimental point, i.e.,

$$\frac{dX_i}{dt} = \frac{X_i - X_{i-\Delta}}{t_i - t_{i-\Delta}} \tag{2}$$

where $\Delta$ is the interval of the experimental data taken into account. In this case, $\Delta = 1$ has been used.

## 2.5. Kinetic Model

The thermochemical decomposition of the biomass can be represented by three main kinetics that correspond to the degradation of hemicellulose, cellulose and lignin. The most commonly used model consists of assuming that the process can be represented by the decomposition reactions of each of these compounds [23,24]. In addition, the decomposition of these compounds can be represented by a number of parallel and independent first-order Arrhenius-type reactions, named pseudocomponents.

Thus, for the adjustment of the DTG curve of each biomass, it has been assumed that the process follows the model that consists of the decomposition of hemicellulose, cellulose and lignin independently, so that the overall kinetics can then be expressed as follows:

$$\frac{dX}{dt} = \frac{dX_H}{dt} + \frac{dX_C}{dt} + \frac{dX_L}{dt} \tag{3}$$

where *H*, *C* and *L* represent the mass fraction of hemicellulose, cellulose and lignin, respectively.

At the same time, the kinetics of each of these fractions can be represented by a set of parallel reactions, expressed in the form:

$$\frac{dX_H}{dt} = \sum_{j=1}^{m_H} \frac{dX_{H_j}}{dt} = -\sum_{j=1}^{m_H} K_{H_j} exp\left(\frac{-E_{H_j}}{RT}\right) X_{H_j} \tag{4}$$

$$\frac{dX_C}{dt} = \sum_{j=1}^{m_C} \frac{dX_{C_j}}{dt} = -\sum_{j=1}^{m_C} K_{C_j} exp\left(\frac{-E_{C_j}}{RT}\right) X_{C_j} \tag{5}$$

$$\frac{dX_L}{dt} = \sum_{j=1}^{m_L} \frac{dX_{L_j}}{dt} = -\sum_{j=1}^{m_L} K_{L_j} exp\left(\frac{-E_{L_j}}{RT}\right) X_{L_j} \tag{6}$$

where *T*: temperature, in K; *R*: ideal gas constant, $8.314 \times 10^{-3}$ kJ (K mol)$^{-1}$; *j*: number of pseudocomponents of the fractions of hemicellulose, cellulose and lignin, which take the values from 1 to the total number of pseudocomponents of each fraction of hemicellulose; cellulose and lignin (*mH*, *mC* and *mL*); $K_{H_j}$, $K_{C_j}$ and $K_{L_j}$: pre-exponential factors of the pseudocomponents of the hemicellulose, cellulose and lignin fractions, expressed in s$^{-1}$ and $E_{H_j}$, $E_{C_j}$ and $E_{L_j}$: activation energies of the pseudocomponents of the hemicellulose, cellulose and lignin fractions, expressed in kJ mol$^{-1}$.

In general, the kinetic equation of each pseudocomponent j, corresponding to fraction *F* (*F* = *H*, *C*, *L*), in a nonisothermal process at constant heating rate $\beta = dT/dt$, is given by

$$\frac{dX_{F_j}}{X_{F_j}} = -\frac{K_{F_j}}{\beta} exp\left(\frac{-E_{F_j}}{RT}\right) dT \tag{7}$$

The integral of the second term can be resolved by using the exponential integral, defined as follows:

$$\int_u^\infty \frac{e^{-u}}{u} du, \ u = \frac{E}{R} \tag{8}$$

Thus, Equation (7), integrated between *To* and *T*, can be expressed in the form

$$X_{F_{j,i}} = X_{F_{j,0}}.exp\left\{-\frac{K_{F_j}}{\beta}\left[T_i.exp\left(\frac{-E_{F_j}}{RT_i}\right) - \int_{E_{F_j}/RT_i}^\infty \frac{exp\left(\frac{-E_{F_j}}{RT}\right)}{T} dT\right]\right\} \tag{9}$$

Therefore, the kinetics of each pseudocomponent depends on three variables: the pre-exponential factor, the activation energy and the initial concentration of the pseudocomponent in the biomass ($X_{F_{j,0}}$).

A restriction that the system must satisfy is that the sum of the mass fractions of all the pseudocomponents must be equal to the mass fraction of all volatiles generated for each instant of time *t* = *i*.

$$X_i = X_{H_i} + X_{C_i} + X_{L_i} = \sum_{j=1}^{m_H} X_{H_{j,i}} + \sum_{j=1}^{m_C} X_{C_{j,i}} + \sum_{j=1}^{m_L} X_{L_{j,i}} \tag{10}$$

Combining Equations (9) and (10) for each instant of discrete time *i* gives a system of equations with $3 \times (m_H + m_C + m_L) - 1$ unknowns, which needs to be solved.

*2.6. Calculation Procedure*

For the calculation of unknown variables, an optimization method based on the minimization by least squares has been used. As an objective function (OF), the square of the errors between the values of the experimental curve and the model has been used for each instant of time $i$, in which the model has been evaluated.

$$O.F. = \sum_{i=1}^{n}\left[\left(\frac{dX}{dt}\right)_{i,exp} - \left(\frac{dX}{dt}\right)_{i,model}\right]^2 \tag{11}$$

The solution has been made with MATLAB using the *lsqcurvefit* command to find the constants that best fit the system of equations. The final solution was obtained when the percentage variation of the OF was less than 0.01% during five consecutive cycles of 200 iterations each ($\Delta OF_5 < 0.01\%$).

The obtained quality of fit (QOF) between the simulated and experimental curves was evaluated with the expression (12).

$$QOF\ (\%) = 100\ x\ \sum_{i=1}^{n}\frac{\sqrt{\left[\left(\frac{dX}{dt}\right)_{i,exp} - \left(\frac{dX}{dt}\right)_{i,model}\right]^2/n}}{\max\left[\left(\frac{dX}{dt}\right)_{i,exp}\right]} \tag{12}$$

where $n$ is the number of experimental points employed (967).

Additionally, the goodness of fit was evaluated by the adjusted R-squared, $R^2_{Adj}$, which represents the response that is explained by the model and was calculated as the ratio between the sum of square of the residuals (SSE) and the total sum of squares (SST) as follows [25]:

$$R^2_{adj} = 1 - \frac{(n-1)xSSE}{(n-(k+1))xSST} = 1 - \frac{(n-1)x\sum_{i=1}^{n}\left[\left(\frac{dX}{dt}\right)_{i,exp} - \left(\frac{dX}{dt}\right)_{i,model}\right]^2}{(n-(k+1))x\sum_{i=1}^{n}\left[\left(\frac{dX}{dt}\right)_{i,exp} - \overline{\left(\frac{dX}{dt}\right)_{i,exp}}\right]^2} \tag{13}$$

where $k$ is the number of variables.

The initial values of the constants were taken after an initial analysis of the kinetics, using as initial seed values the restrictions on the concentrations of the hemicellulose, cellulose and lignin fractions obtained from the literature review (Table 1).

**Table 1.** Literature references of the main lignocellulosic fraction compositions related to used biomasses.

| Biomass | Ref. | Hemicellulose, wt.% | Cellulose, wt.% | Lignin, wt.% | Extractives, wt.% | Ash, wt.% |
|---|---|---|---|---|---|---|
| Pine bark | [26] [a] | 25.0 | 19.0 | 38.0 | 18.0 | |
| Spruce bark | [24] [a] | 27.0 | 42.0 | 26.0 | | |
| | [27] [a] | 24.3 | 41.0 | 30.0 | | |
| | [28] [a] | 21.2 | 50.8 | 27.5 | | |
| | [26] [a] | 28.0 | 22.0 | 31.0 | 19.0 | |
| Poplar | [17] [b] | 26.0 | 50.0 | 24.0 | | |
| | [19] [a] | 28.0 | 43.0 | 25.0 | 5 | |
| | [29] [a] | 18.0–26.6 | 46.5–52.0 | 16.0–25.9 | | |
| | [3] [b] | 22.0 | 49.0 | 28.0 | | |
| | [30] [a] | 24.0 | 49.0 | 20.0 | 5.9 | 1.0 |
| Willow | [30] [a] | 16.7 | 41.7 | 29.3 | 9.7 | 2.5 |
| Wheat straw | [30] [a,c] | 24.6 | 39.2 | 17.0 | | |
| | [28] [a] | 29.0 | 38.0 | 15.0 | | |
| | [28] [a] | 39.1 | 28.8 | 18.6 | | |
| | [30] [a] | 25.0 | 37.5 | 20.2 | 4.0 | 3.7 |

[a] By chemical methods, [b] by thermogravimetric analysis (TGA) and [c] cellulose as glucan and hemicellulose as xylan.

The decision tree of the calculation process is as Figure 1:

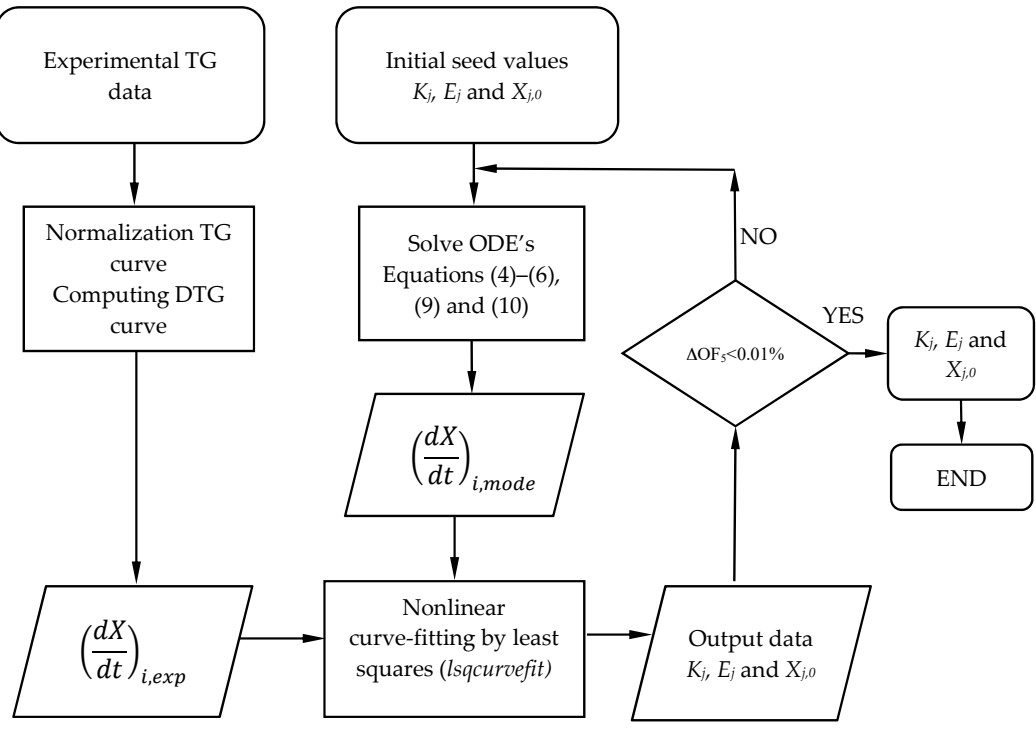

**Figure 1.** Decision tree of the calculation procedure.

## 3. Results and Discussion

### 3.1. Analytical Method

The lignocellulosic biomass wt.% composition was determined by chemical methods by the VTT and TECNALIA laboratories; the detailed procedure was described in [31]. Biomasses were previously sampled and prepared through TAPPI T257 and then conditioned through TAPPI 264. Table 2 includes the analytical results obtained.

**Table 2.** Composition by chemical methods for the raw biomasses (wt.%, dry basis).

| Biomass Component | Analysis Method | Pine Bark | Spruce Bark | Poplar | Willow | Wheat Straw |
|---|---|---|---|---|---|---|
| Hemicellulose | TAPPI T249 | 18.30 | 13.90 | 21.70 | 22.60 | 23.80 |
| Cellulose | TAPPI T249 | 21.90 | 29.70 | 42.70 | 44.30 | 37.50 |
| Lignin | TAPPI T222 | 40.70 | 45.10 | 26.90 | 25.10 | 20.50 |
| Extractives | Internal Method | 15.20 | | 4.40 | 8.00 | 15.70 |
| | TAPPI 204 | | 4.90 | | | |
| Ash | XP CEN/TS 14775 | 2.80 | | 2.80 | 2.30 | 8.30 |
| | TAPPI 211 | | 5.22 | | | |

The results obtained in Table 2 are in-line with the results obtained by other researchers [32]. According to the literature, the softwood bark composition corresponds to a cellulose content of 18–38%, the hemicellulose content is 15–33% and the lignin content is 30–60%. For hardwood biomasses, the cellulose content is 43–47%, the hemicellulose content is 25–35% and the lignin content is 16–24%. Finally, the composition of herbaceous biomass, such as cereal straw, is 33–38% cellulose, 26–32% hemicellulose and 17–19% lignin.

Therefore, according to the literature review [32,33] and the analyses carried out (Table 2), softwood bark has higher lignin content than hardwood and agricultural biomasses. On the other hand, hardwood has a higher cellulose content than the rest of the biomasses analyzed.

It also should be noted that, during the thermogravimetric analysis (TGA), it is possible to differentiate the biomass into its three main lignocellulosic fractions, but it is not possible to distinguish the extractives from the other fractions. Extractives are a group of compounds that can be obtained from the biomass using organic solvents, such as benzene, alcohol or water [34]. The main components of the lipophilic extracts are triglycerides, fatty acids, resin acids, sterile esters and sterols and of hydrophilic extracts are lignin [35]. The extractives thermally degrade in the temperature range of 200–400 °C, which falls within the range in which hemicellulose and cellulose and, also, lignin is degraded. For this reason, in order to get comparable results with those obtained by the TGA method, the analytical data are expressed in weight % on a dry and ash and extractives-free basis.

## 3.2. Devolatilization Behavior

The performance of the DTG curves shows similar behavior (Figure 2). At first sight, two large peaks can be observed in all of them: the first one appears from room temperature to about 150 °C and corresponds to the loss of moisture. At temperatures exceeding 150 °C, degradation of lignocellulosic compounds begins [30,32,36,37]. The second large peak is located in the range of temperature between 250 and 380 °C and corresponds to the degradation of cellulose. Two other peaks, which are more or less perceptible depending on the type of biomass, can be seen overlapping the cellulose peak. Thus, at temperatures between 200 and 300 °C, the degradation of hemicellulose occurs, which proves a deformation of the cellulose peak in that temperature range. Finally, lignin is the component with the most complex structure, and its decomposition range is the widest, occurring from 200 °C to the final temperature of the analysis. The degradation of lignin is more significant near the 400 °C zone, where a small peak can be observed that overlaps with the end of the cellulose degradation.

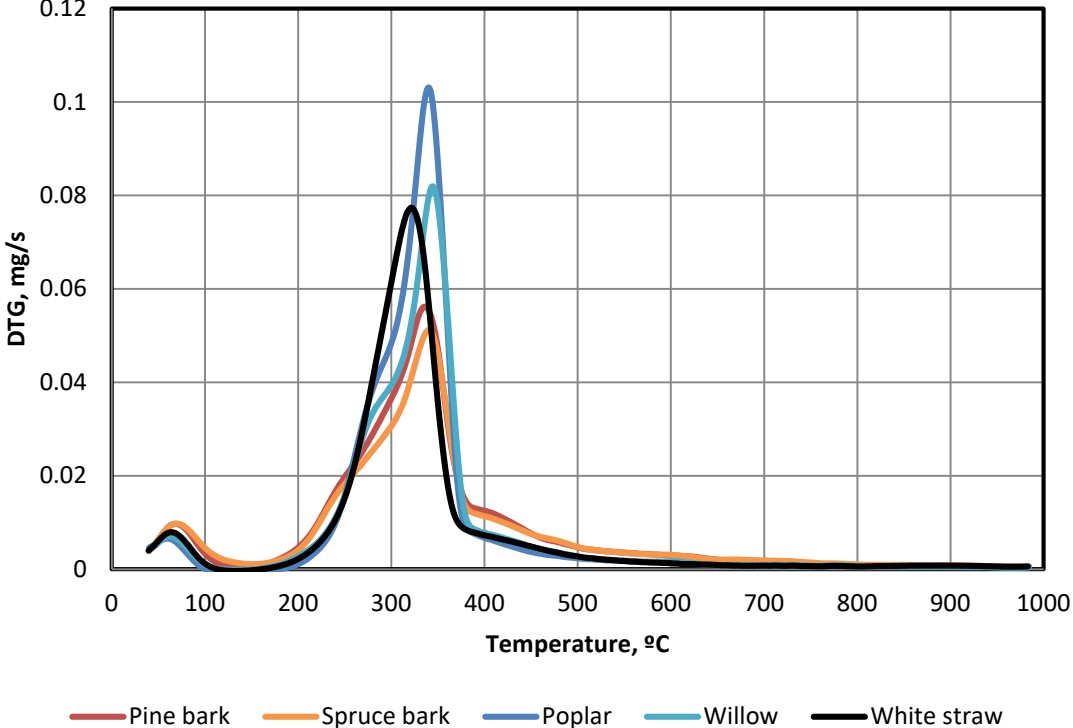

**Figure 2.** DTG curves comparison.

In relation to the development of each type of biomass, it is observed that pine bark and spruce bark have very similar development patterns. Both barks, as compared to the rest of the biomasses (poplar, willow and white straw), have a higher peak near 400 °C corresponding to the degradation of lignin and a lower peak height corresponding to the degradation of cellulose (~350 °C) and hemicellulose (~300 °C). Therefore, these softwood barks have a higher lignin content and lower cellulose and hemicellulose contents, as compared to other biomasses (Table 2). It is also observed that these two biomasses have the lowest DTG area, so they are the ones that release the least amounts of total volatiles.

On the other hand, willow and poplar show very similar behaviors, which indicates that their compositions will be very similar. Both biomasses present a greater generation of volatiles in the cellulose degradation zone. This is in agreement with the fact that both biomasses have higher cellulose contents and lower lignin contents compared to the rest of the biomasses analyzed (Table 2).

Finally, wheat straw presents a single peak in the degradation zone of hemicellulose and cellulose and is slightly displaced to the low temperature zone. This suggests a higher hemicellulose content, while the evolution of the lignin content is very similar to that of poplar and willow.

### 3.3. TGA-PKM Method

The first step was to determine the minimum number of pseudocomponents needed to adequately represent the evolution of each of the three main lignocellulosic fractions and all volatiles generated during the thermal degradation process.

This analysis was carried out by means of an initial kinetic analysis, in which a division of the DTG was established according to the degradation temperatures of the three main constituents of the biomass (hemicellulose, cellulose and lignin), in addition to water. Each of these regions was initially attributed a single pseudocomponent; then, the number of pseudocomponents was gradually increased, until an adequate performance of the evolution of the volatiles was achieved. The minimum numbers of pseudocomponents necessary for the quantifications of each fraction are shown in the Table 3. The use of a larger number of pseudocomponents could induce overfitting.

**Table 3.** Minimum number of components for each biomass fraction.

| Component | Temperature Range, °C | Number of Pseudocomponents |
|---|---|---|
| Water | 25–150 | 1 |
| Hemicellulose | 200–350 | 2 |
| Cellulose | 250–400 | 1 |
| Lignin | 150–1000 | 3 |

The next step was to determine the minimum number of heating rates needed to achieve the objective of quantifying the main biomass fractions. The use of three or more heating rates while reducing the effect of kinetic compensation and improving the accuracy of kinetic parameters requires the use of significantly different heating rates, which involves, in practice, the use of higher heating rates. However, higher heating rates worsen the separation of lignocellulosic fractions, making their identification more difficult. Additionally, the use of various heating rates for the quantification of the lignocellulosic fractions is more time-consuming.

Therefore, a low heating rate achieves a better separation of the degraded compounds and is less time-consuming. This is the reason why a single heating rate of 5 °C min$^{-1}$ has been employed in the determination of the main lignocellulosic fractions. However, a validation of the method using three heating rates has been carried out and is reported in Section 3.4.

To improve the accuracy of the kinetic parameters, it was found that the use of upper and lower limits of the kinetic parameters (Tables 4 and 5) was necessary, not only to ensure adequate values of the pre-exponential and activation energy but, also, to provide adequate seed values for the determination of the hemicellulose, cellulose and lignin fractions.

**Table 4.** Upper bonds of the pseudocomponents (PC).

| Kinetic Parameters | PC 2 | PC 3 | PC 4 | PC 5 | PC 6 | PC 7 |
|---|---|---|---|---|---|---|
| $K$ (s$^{-1}$) | $1.00 \times 10^9$ | $1.50 \times 10^5$ | $2.40 \times 10^{15}$ | $5.00 \times 10^1$ | 3.00 | 1.80 |
| $E$ (kJ mol$^{-1}$) | 120.00 | 80.00 | 240.00 | 60.00 | 60.00 | 68.00 |
| $X_{j,0}$ (wt.%) | 50.00 | 50.00 | 60.00 | 60.00 | 20.00 | - |

**Table 5.** Lower bonds of the pseudocomponents (PC).

| Kinetic Parameters | PC 2 | PC 3 | PC 4 | PC 5 | PC 6 | PC 7 |
|---|---|---|---|---|---|---|
| $K$ (s$^{-1}$) | $7.00 \times 10^8$ | $1.40 \times 10^5$ | $1.50 \times 10^{15}$ | $4.00 \times 10^7$ | 2.30 | $1.00 \times 10^{-1}$ |
| $E$ (kJ mol$^{-1}$) | 100.00 | 70.00 | 160.00 | 55.00 | 45.00 | 50.00 |
| $X_{j,0}$ (wt.%) | 0.1 | 1 | 5.00 | 15.00 | 0.10 | - |

Finally, taking into account the above procedure, the values of the kinetic parameters of each pseudocomponent were calculated by the TGA-PKM method and are summarized in Table 6.

Overall, the results obtained in this study are in reasonable ranges when compared to the results corresponding to the kinetics of other biomasses published, as can be seen in Table 7.

**Table 6.** Kinetic parameters of the pseudocomponents.

| Biomass | Kinetic Parameters | Water PC 1 | Hemicellulose PC 2 | Hemicellulose PC 3 | Cellulose PC 4 | Cellulose PC 5 | Lignin PC 6 | Lignin PC 7 |
|---|---|---|---|---|---|---|---|---|
| Pine bark | $K$ (s$^{-1}$) | $9.14 \times 10^4$ | $6.00 \times 10^8$ | $1.50 \times 10^5$ | $1.69 \times 10^{15}$ | $5.00 \times 10$ | 2.44 | $9.83 \times 10^{-1}$ |
| | $E$ (kJ mol$^{-1}$) | 48.58 | 120.00 | 75.60 | 204.74 | 55.00 | 50.20 | 61.31 |
| | $X_{j,0}$ (wt.%) | 6.81 | 14.71 | 7.00 | 24.66 | 27.68 | 13.16 | 5.98 |
| Spruce bark | $K$ (s$^{-1}$) | $4.52 \times 10^3$ | $7.00 \times 10^8$ | $1.42 \times 10^5$ | $1.51 \times 10^{15}$ | $5.00 \times 10$ | 2.33 | 1.11 |
| | $E$ (kJ mol$^{-1}$) | 40.74 | 119.99 | 74.93 | 205.17 | 55.00 | 48.79 | 60.95 |
| | $X_{j,0}$ (wt.%) | 8.25 | 14.04 | 6.56 | 24.53 | 24.61 | 14.18 | 7.82 |
| Poplar | $K$ (s$^{-1}$) | $7.22 \times 10^5$ | $6.00 \times 10^8$ | $1.40 \times 10^5$ | $2.30 \times 10^{15}$ | $4.81 \times 10$ | 2.79 | $5.07 \times 10^{-1}$ |
| | $E$ (kJ mol$^{-1}$) | 52.44 | 120.00 | 80.00 | 207.39 | 55.00 | 53.22 | 59.71 |
| | $X_{j,0}$ (wt.%) | 3.81 | 21.72 | 1.00 | 51.85 | 15.34 | 4.24 | 2.04 |
| Willow | $K$ (s$^{-1}$) | $2.98 \times 10^5$ | $6.00 \times 10^8$ | $1.47 \times 10^5$ | $1.94 \times 10^{15}$ | $5.00 \times 10$ | 2.60 | $8.77 \times 10^{-1}$ |
| | $E$ (kJ mol$^{-1}$) | 50.69 | 120.00 | 74.88 | 208.03 | 55.00 | 49.43 | 62.61 |
| | $X_{j,0}$ (wt.%) | 4.69 | 21.91 | 1.91 | 44.33 | 15.00 | 8.25 | 3.91 |
| Wheat straw | $K$ (s$^{-1}$) | $8.23 \times 10^5$ | $6.00 \times 10^8$ | $1.50 \times 10^5$ | $1.51 \times 10^{15}$ | $5.00 \times 10$ | 3.00 | $1.62 \times 10^{-1}$ |
| | $E$ (kJ mol$^{-1}$) | 53.65 | 120.00 | 71.38 | 200.64 | 55.00 | 51.08 | 52.16 |
| | $X_{j,0}$ (wt.%) | 5.18 | 24.16 | 1.00 | 39.51 | 19.73 | 6.90 | 3.51 |

**Table 7.** Kinetic parameters from other studies.

| Component | Temperature, °C | E, kJ mol$^{-1}$ | K, min$^{-1}$ | Reference |
|---|---|---|---|---|
| Hemicellulose | 200–350 | 127.00 | $9.5 \times 10^{10}$ | [38] |
| | | 83.20–96.40 | $4.55 \times 10^6$–$1.57 \times 10^8$ | [39] |
| Cellulose | 300–340 | 227.02 | $3.36 \times 10^{18}$ | [37] |
| | | 239.70–325.00 | $16.30 \times 10^{19}$–$3.62 \times 10^{26}$ | [39] |
| Lignin | 220–380 | 7.80 | $2.96 \times 10^{-3}$ | [37] |
| | 25–900 | 47.90–54.50 | $6.80 \times 10^2$–$6.60 \times 10^4$ | [17] |
| | 160–680 | 25.20 | $4.70 \times 10^2$ | [18] |
| | | 20.00–29.10 | $5.35 \times 10$–3.18 | [39] |

As shown in Tables 6 and 7, cellulose is the compound with the highest activation energies. This is attributed to the fact that the cellulose is a very long polymer of glucose units without any branches [18], while hemicellulose has a random branched amorphous structure that gives a lower activation energy; this is the reason why hemicellulose decomposes more easily in a lower temperature range [38].

Lignin has a very complex structure composed of three kinds of heavily crosslinked phenylpropane structures [18]. Additionally, it is observed that the activation energy is lower than for hemicellulose and cellulose, which indicates that its thermal degradation is easier. However, it presents much lower values of pre-exponential factors that cause a lower reaction rate; this fact is reflected in the wide range of temperatures in which its degradation takes place and in the high temperature required to reach a complete degradation.

In addition, Figures 3–7 show the fit of the model to the DTG experimental data, as well as the contribution of the different pseudocomponents to the model. In all the figures, it can be seen that a good fit is achieved between the global model, obtained as the envelope resulting from the sum of the seven pseudocomponents, and the experimental DTG curve.

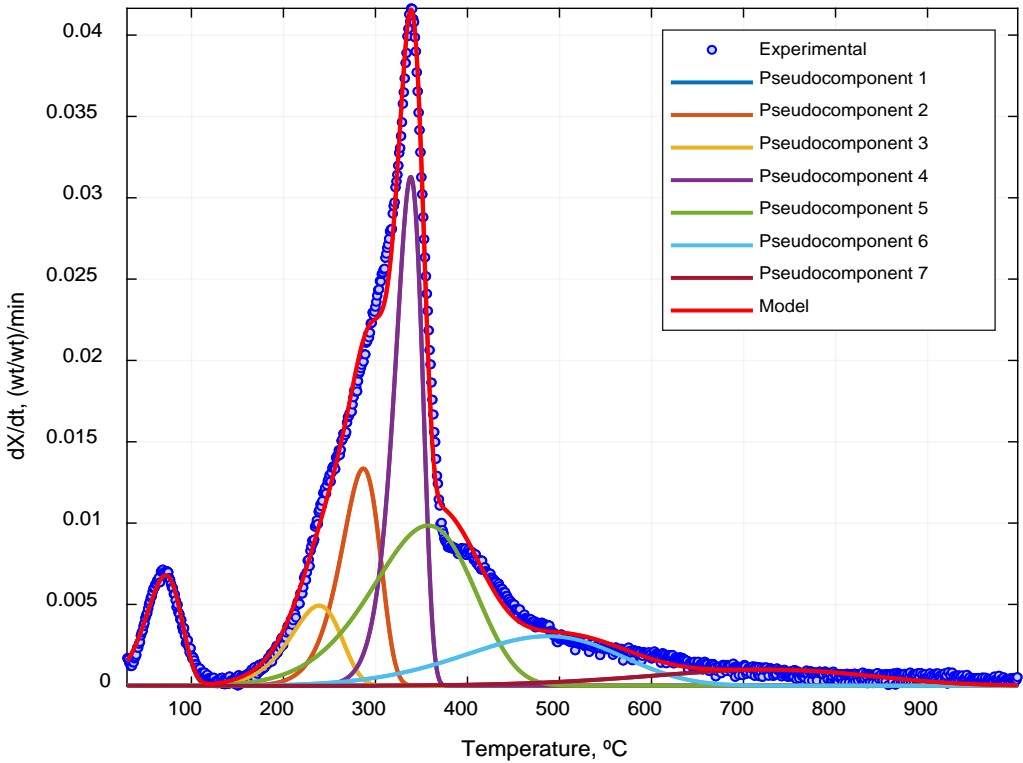

**Figure 3.** Model fitted to the experimental pine bark DTG curve.

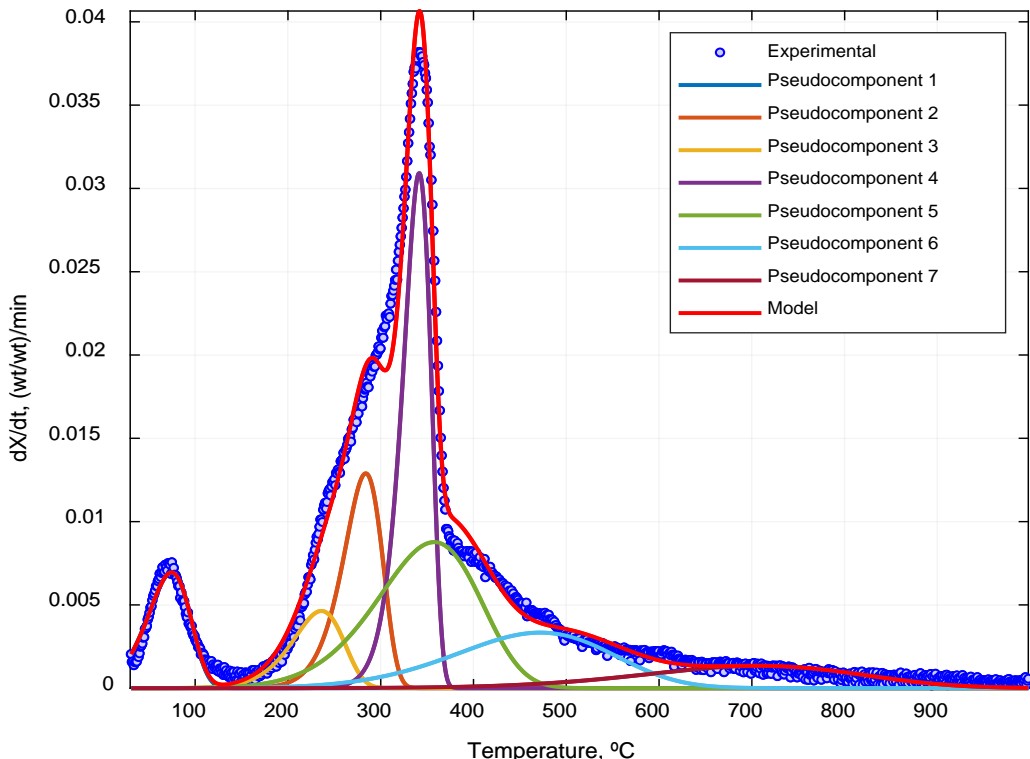

**Figure 4.** Model fitted to the experimental spruce bark DTG curve.

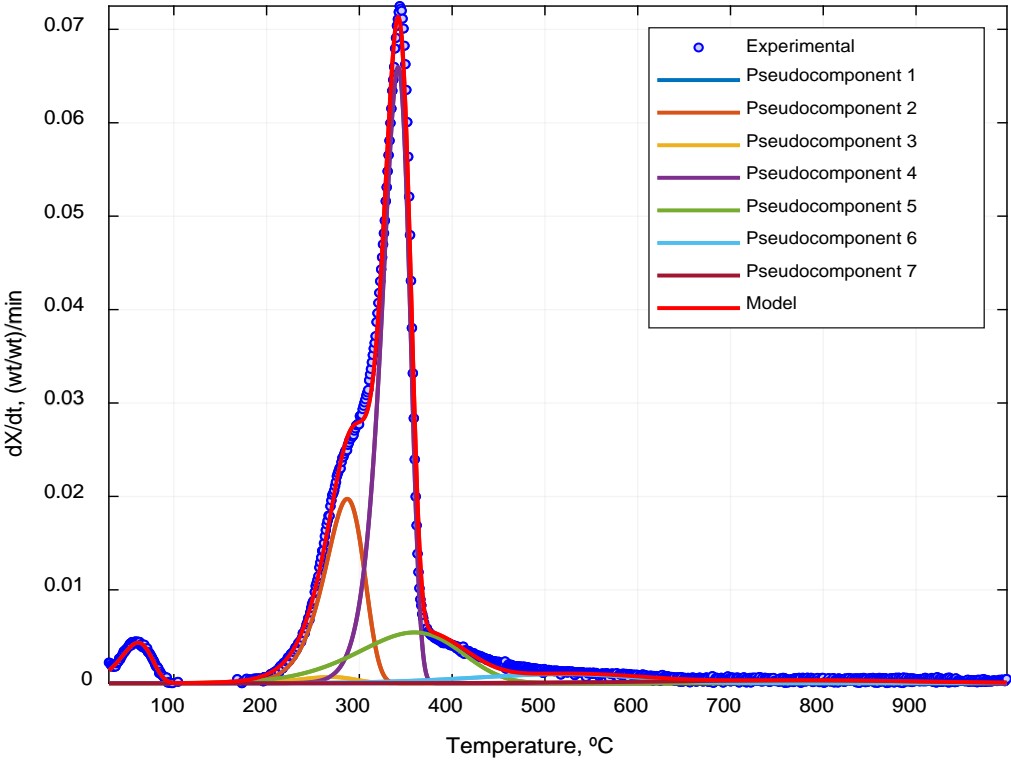

**Figure 5.** Model fitted to the experimental poplar DTG curve.

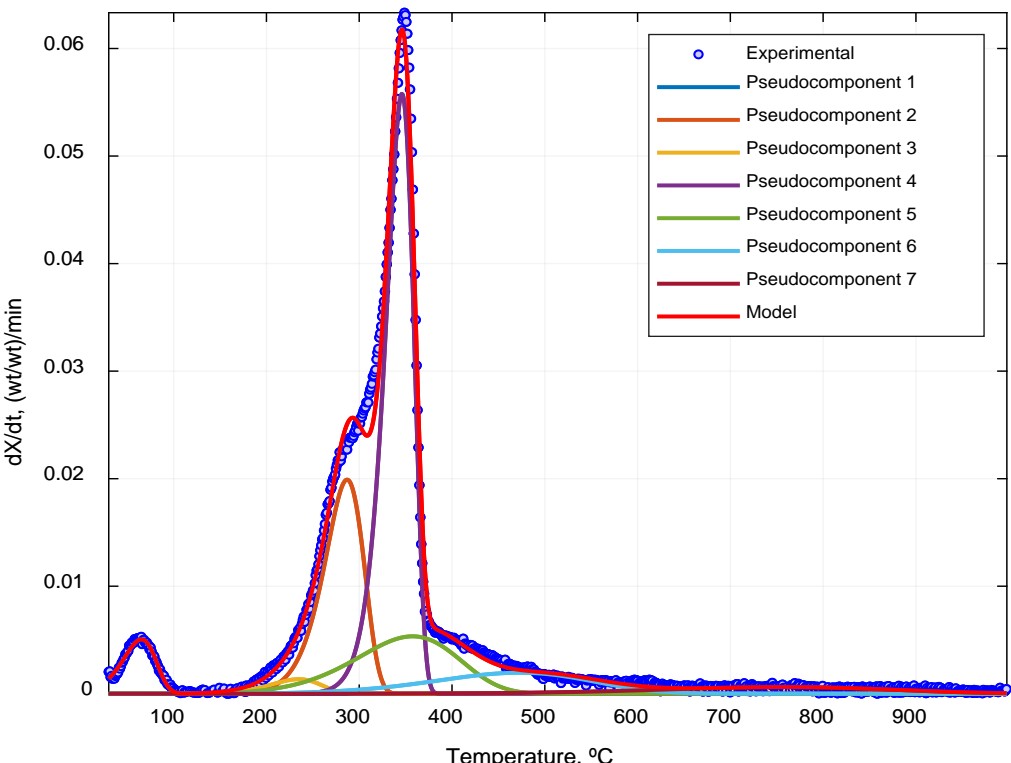

**Figure 6.** Model fitted to the experimental willow DTG curve.

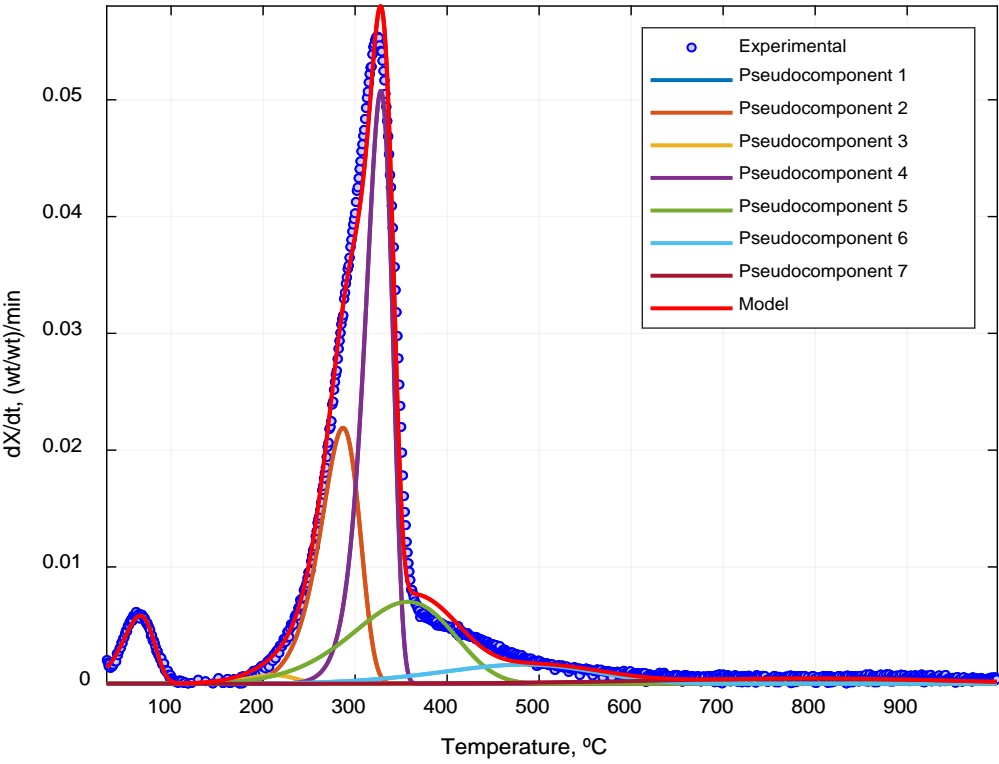

**Figure 7.** Model fitted to the experimental wheat straw DTG curve.

By comparison, between the kinetic constants in Table 6 and Figures 3–7, it can be seen that low activation energy leads to a reaction in the low temperature zone and vice versa. With respect to the pre-exponential factor, low values cause the reaction rate to be slower and to take place over a

wider temperature range, which is characteristic of the lignin pseudocomponents. On the contrary, high values of the pre-exponential factor increase the reaction rate, leading to a narrower temperature range, which is characteristic of cellulose, for example.

On the other hand, at the same activation energy, a higher pre-exponential factor causes the reaction to take place in the high temperature zone. For example, there are lignin pseudocomponents with a similar activation energy as hemicellulose pseudocomponents (Table 6) but with much lower pre-exponential factors, which cause the reaction to take place at higher temperatures.

The quality of the fit expressed as $R^2$ and QOF% can be observed in Table 8.

**Table 8.** Quality of the fit expressed as $R^2_{Adj}$ and QOF%.

| Biomass | QOF% | $R^2_{Adj}$ |
|---------|------|-------------|
| Pine bark | 1.51 | 0.9939 |
| Spruce bark | 1.94 | 0.9905 |
| Poplar | 1.09 | 0.9960 |
| Willow | 1.42 | 0.9933 |
| Wheat straw | 1.79 | 0.9921 |

In addition, Figures 8–12 show the fit of the global model to the TG experimental data. The TG curve model has been obtained simultaneously with the DTG curve model by solving Equations (9) and (10). As can be seen, the TG curve model achieves good results not only with respect to the model fitting to the experimental TG curve along the operating temperature but, also, with respect to the final value.

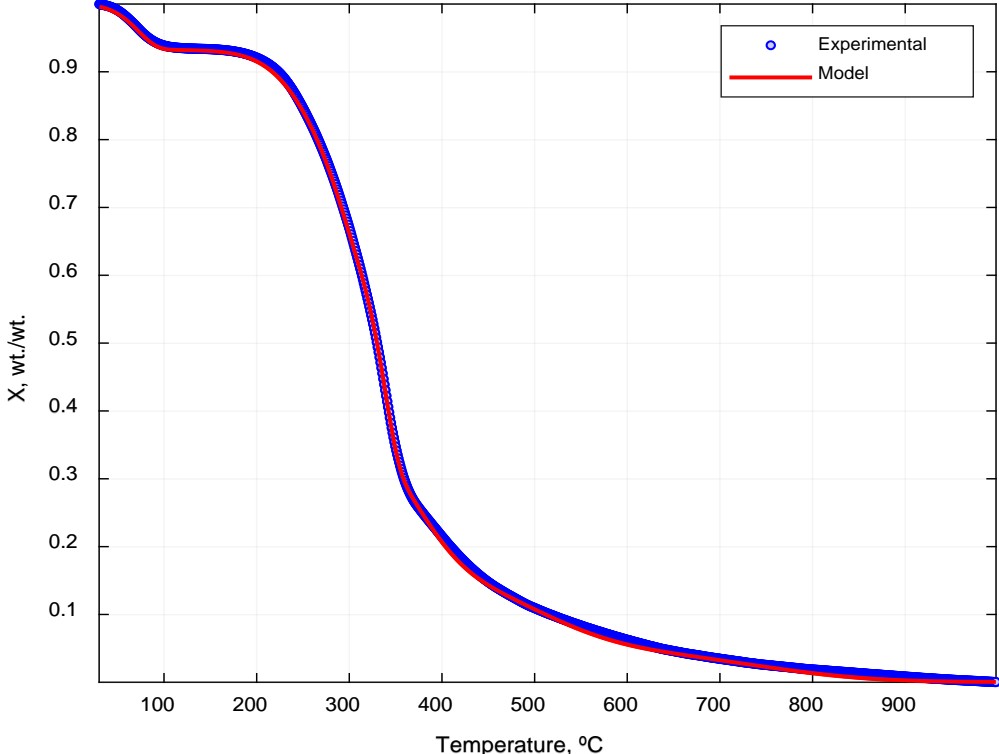

**Figure 8.** Model fitted to the experimental pine bark TG curve.

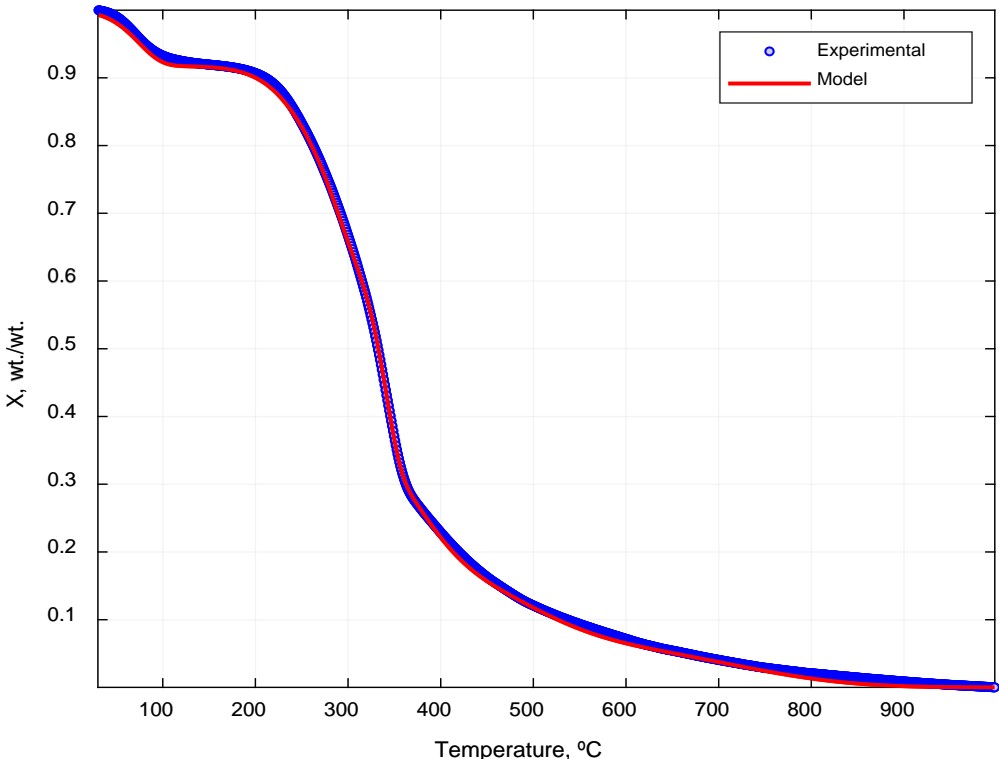

**Figure 9.** Model fitted to the experimental spruce bark TG curve.

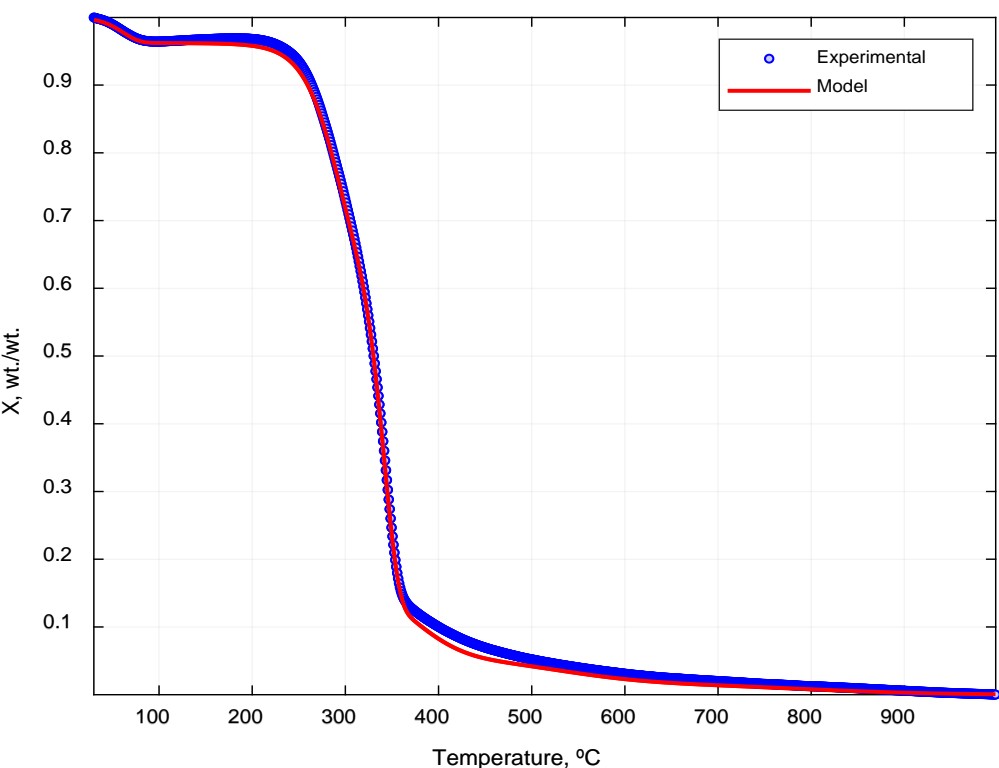

**Figure 10.** Model fitted to the experimental poplar TG curve.

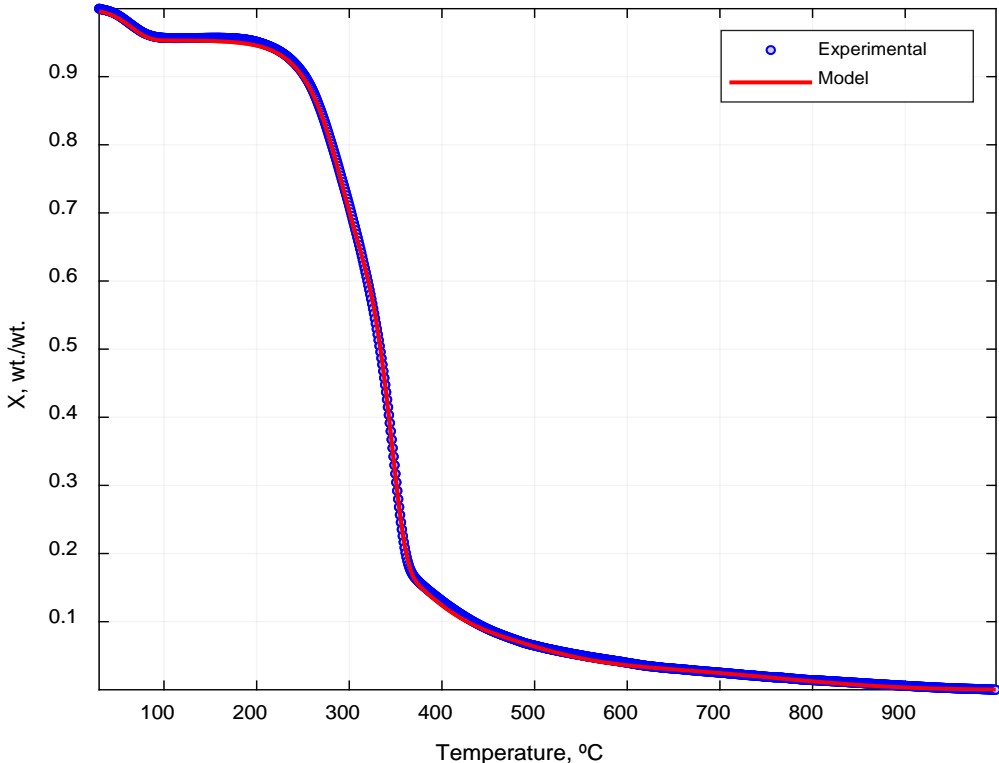

**Figure 11.** Model fitted to the experimental willow TG curve.

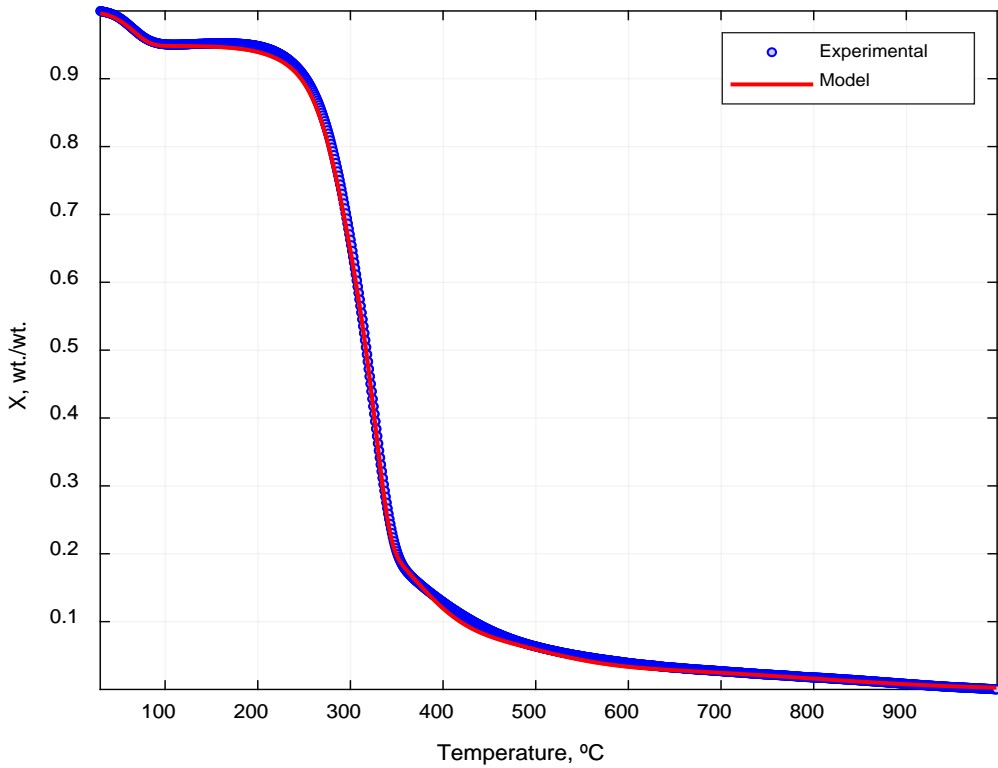

**Figure 12.** Model fitted to the experimental wheat straw TG curve.

Table 9 shows the comparison between the analytical composition and the data obtained with the TGA-PKM method. As can be seen, there is a good agreement between the data obtained through analytical procedures and the TGA-PKM model. This indicates that the new method can be used to

have a good estimation of the content of the main lignocellulosic fractions of the analyzed biomasses without the need to carry out complex extraction and purification chemical treatments.

**Table 9.** Comparison between the analytical and thermogravimetric analysis-pseudocomponent kinetic model (TGA-PKM) results.

| Biomass | Component | Analytical Method wt.%, Dry, Ash and Extractives-Free Basis | TGA-PKM Method wt.%, Dry, Ash and Extractives-Free Basis | Error, wt.% |
|---|---|---|---|---|
| Poplar | Hemicellulose | 23.77 | 23.62 | −0.15 |
| | Cellulose | 46.77 | 53.90 | 7.14 |
| | Lignin | 29.46 | 22.48 | −6.99 |
| Willow | Hemicellulose | 24.57 | 25.00 | 0.43 |
| | Cellulose | 48.15 | 46.51 | −1.64 |
| | Lignin | 27.28 | 28.49 | 1.21 |
| Wheat straw | Hemicellulose | 29.10 | 26.54 | −2.56 |
| | Cellulose | 45.84 | 41.67 | −4.18 |
| | Lignin | 25.06 | 31.80 | 6.73 |
| Spruce Bark | Hemicellulose | 15.67 | 22.45 | 6.78 |
| | Cellulose | 33.48 | 26.74 | −6.74 |
| | Lignin | 50.85 | 50.81 | −0.04 |
| Pine bark | Hemicellulose | 22.62 | 23.30 | 0.68 |
| | Cellulose | 27.07 | 26.47 | −0.61 |
| | Lignin | 50.31 | 50.24 | −0.07 |

The following error ranges are obtained between the values measured analytically and those measured by the TGA-PKM method for each of the main lignocellulosic fractions: hemicellulose (−2.56–6.78), cellulose (−6.74–7.14) and lignin (−6.99–6.73). The level of accuracy achieved is considered suitable, taking into account that it is within the error range of the chemical methods. For example, Korpinen et al. found that the determination of lignin by different chemical methods can be as high as 10 wt.% [39]; Ioelovich [40] also determined a difference of 4 wt.% between the TAPPI and NERL methods in determination of the cellulose content. In this way, the TGA-PKM method allows to obtain a fast estimation of the contents of the main lignocellulosic fractions within the ranges that would be obtained by a chemical analysis.

*3.4. Validation of the TGA-PKM Method*

In order to check the validity of the method, an additional fit of the poplar biomass devolatilization was performed using, simultaneously, three heating rates: 3, 5 and 10 °C min$^{-1}$ datasets.

Figure 13 shows the graphical results by fitting the model to the DTG and TG curves for each heating rate.

Additionally, the quality of the fit achieved for each heating rate and for the global fit are summarized in Table 10, where QOF% and $R^2_{Adj}$ are shown for each heating rate dataset and for the three heating rates simultaneously.

**Table 10.** Quality of the fit for each dataset.

| Quality of the Fit | 3 °C min$^{-1}$ | 5 °C min$^{-1}$ | 10 °C min$^{-1}$ | Global |
|---|---|---|---|---|
| QOF% | 1.16 | 1.58 | 0.96 | 1.35 |
| $R^2_{Adj}$ | 0.9957 | 0.9917 | 0.9971 | 0.9959 |

The results obtained (Figure 13 and Table 10) indicate that the quality of the fit obtained is very satisfactory, since the model is capable of representing the evolution of the devolatilization process when different heating rates are used.

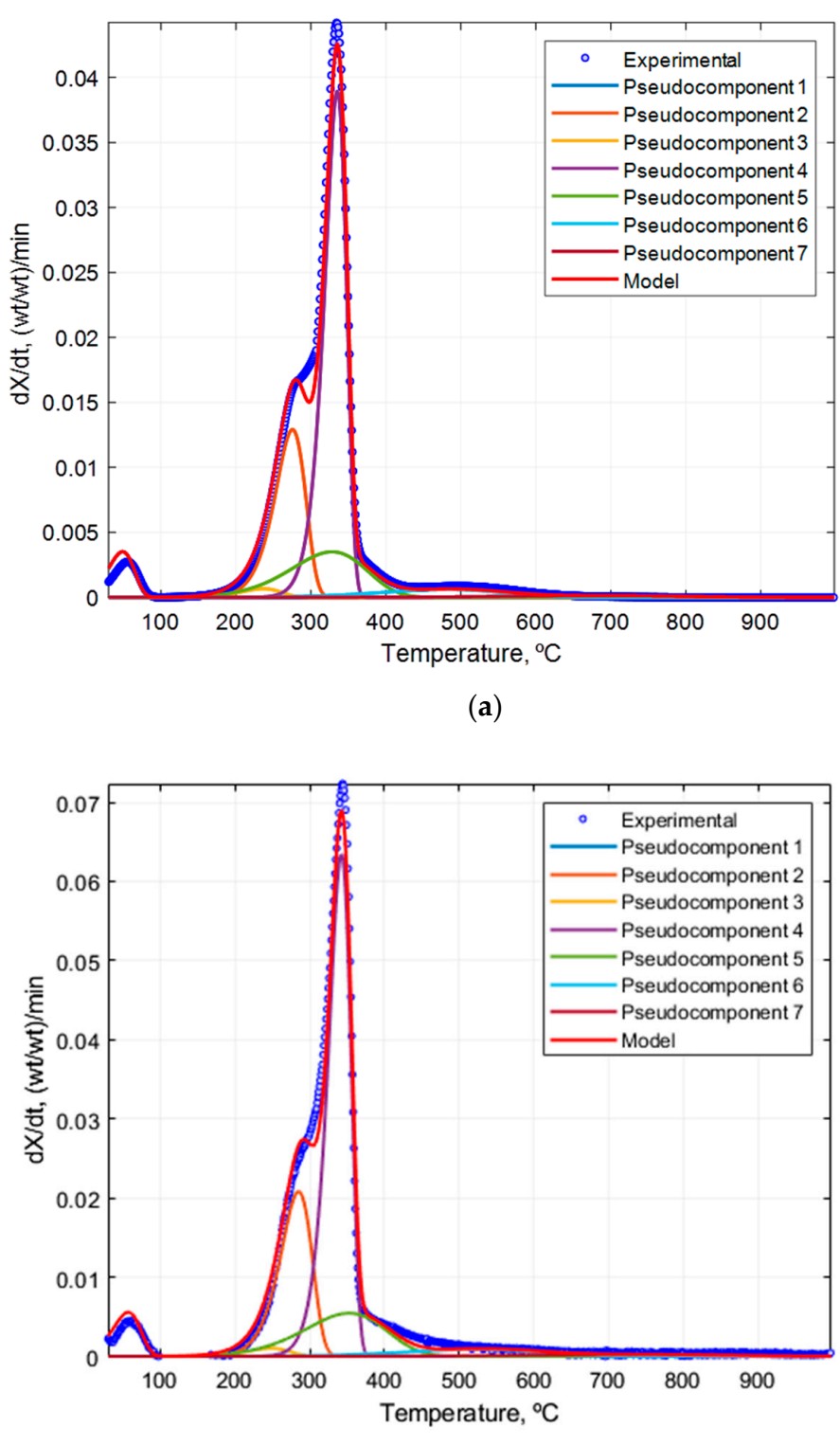

(**a**)

(**b**)

**Figure 13.** *Cont.*

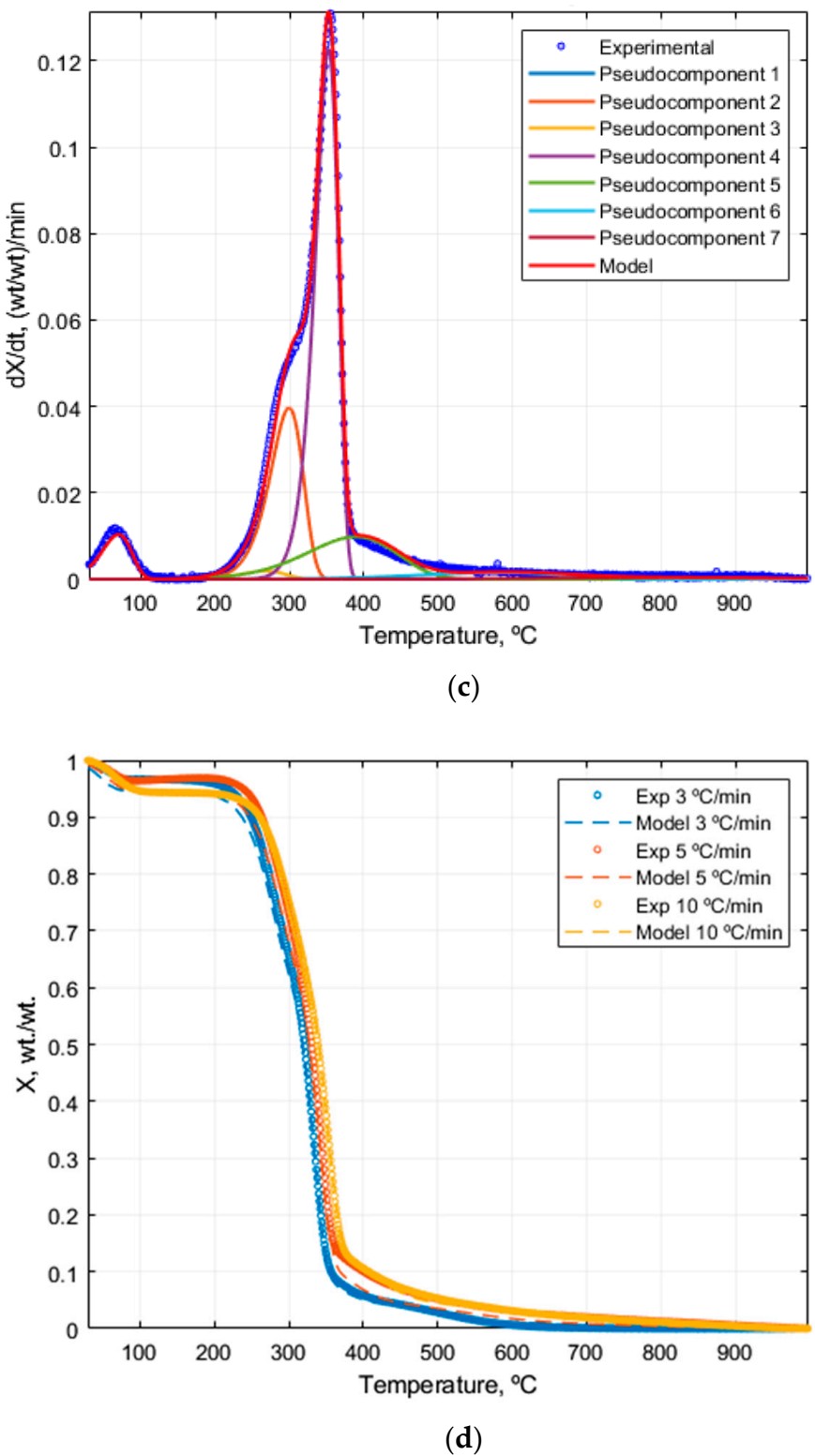

**Figure 13.** Model fitted to the experimental poplar DTG and TG curves: (**a**) DTG at 3 °C min−1, (**b**) 5 °C min−1 and (**c**) 10 °C min−1. (**d**) TG at the three heating rates.

Table 11 shows the kinetic parameters by fitting the model using a single heating rate and three heating rates simultaneously. The obtained results by both datasets are very similar. For example, the activation energy obtained is identical for almost all the pseudocomponents, and only pseudocomponents 3 and 7 have a relative standard deviation of 4%.

**Table 11.** Kinetic parameters of each pseudocomponents calculated using a single heating rate and three simultaneous heating rates.

| Number of Heating Rates | Kinetic Parameters | Hemicellulose PC 2 | PC 3 | Cellulose PC 4 | PC 5 | Lignin PC 6 | PC 7 |
|---|---|---|---|---|---|---|---|
| Single heating rate | K (s$^{-1}$) | $6.00 \times 10^8$ | $1.40 \times 10^5$ | $2.30 \times 10^{15}$ | $4.81 \times 10^1$ | $2.79 \times 10^0$ | $5.07 \times 10^{-1}$ |
| | E (kJ mol$^{-1}$) | 120.00 | 80.00 | 207.39 | 55.00 | 53.22 | 59.71 |
| | X$_{j,0}$ (wt.%) | 21.72 | 1.00 | 51.85 | 15.34 | 4.24 | 2.04 |
| Three simultaneous heating rates | K (s$^{-1}$) | $6.56 \times 10^8$ | $1.50 \times 10^5$ | $2.23 \times 10^{15}$ | $5.49 \times 10^1$ | $2.79 \times 10^0$ | $5.57 \times 10^{-1}$ |
| | E (kJ mol$^{-1}$) | 119.98 | 77.18 | 207.48 | 55.00 | 53.99 | 57.48 |
| | X$_{j,0}$ (wt.%) | 22.78 | 1.46 | 49.97 | 15.08 | 3.88 | 1.51 |

Finally, the results obtained in the determination of the lignocellulosic fractions are shown in Table 12. The results obtained with a single heating rate are comparable to those obtained with three heating rates, because the deviation between the results calculated by the model and by the analytical method are of the same order when a single heating rate or three heating rates are considered. However, slightly better results are achieved if a single heating rate of 5 °C min$^{-1}$ is used, but mainly, it requires considerably less analysis time, which justifies the use of a single heating rate.

**Table 12.** Comparison between TGA-PKM results using three simultaneous heating rates and the analytical method.

| Biomass | Component | Analytical Method Wt.%, Dry, Ash and Extractives-Free Basis | TGA-PKM Method (Three Simultaneous Heating Rates) wt.%, Dry, Ash and Extractives-Free Basis | Error, wt.% |
|---|---|---|---|---|
| Poplar | Hemicellulose | 23.77 | 25.60 | 1.83 |
| | Cellulose | 46.77 | 52.78 | 6.01 |
| | Lignin | 29.46 | 21.62 | −7.85 |

## 4. Conclusions

Five lignocellulosic samples have been characterized by the TGA-PKM experimental protocol, covering different types of woody and herbaceous biomasses from both forest and agricultural origins (spruce bark, pine bark, poplar, willow and wheat straw).

The TGA-PKM method developed allows the determination of the main lignocellulosic fractions of biomasses without the need to use long and complex chemical methods; e.g., TAPPI methods T222 and T249 require several long successive steps (hydrolysis, extraction, filtration, neutralization, reduction, etc.) [41], which may require several days of work in the laboratory, while the new method may be performed in a few hours. Thus, it would be possible to reduce the cost of analysis and processing time by 80–90%.

The accuracy of the TGA-PKM method was tested and proved to be significantly good and consistent within the order of magnitude of the standard analytical methods to determine the contents of the main lignocellulosic fractions.

**Author Contributions:** Conceptualization, D.D. and A.U.; methodology, D.D. and A.U.; software, D.D. and A.U.; validation, D.D., A.U., R.P.; formal analysis, D.D., A.U., A.B. and T.T.; investigation, D.D. and A.U.; resources, A.B., T.T. and R.P.; data curation, D.D., A.U.; writing (original draft preparation), D.D., A.U.; writing (review and editing), R.P, A.B. and T.T.; visualization, D.D., A.U., R.P., A.B. and T.T.; supervision, D.D., A.U., R.P., A.B. and T.T.; project administration, A.B; funding acquisition, A.B., T.T. and R.P. All authors have read and agreed to the published version of the manuscript.

**Funding:** This research was funded by the European Union's Horizon 2020 research and innovation programme under grant agreement No 723670, with the title "Systemic approach to reduce energy demand and CO$_2$ emissions of processes that transform agroforestry waste into high added value products (REHAP)".

**Acknowledgments:** The authors would like to thank María González Martínez from IMT Mines Albi (Université de Tolousse) for her technical contribution and support.

**Conflicts of Interest:** The authors declare no conflict of interest.

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
