# Peer review of "Determination of Hemicellulose, Cellulose, and Lignin Content in Different Types of Biomasses by Thermogravimetric Analysis and Pseudocomponent Kinetic Model (TGA-PKM Method)"

_processes, doi:10.3390/pr8091048_

Round 1

Reviewer 1 Report

The manuscript of David Díez et al is undoubtedly of interest for the proposed original method for determining the composition of biomass. However, I would advise authors to pay attention to the structural component of the samples. Indeed, for an amorphous sample and a sample containing a crystalline fraction, the temperature of the onset of pyrolysis, its activation energy, etc. will be different. In my opinion, the work will benefit in quality if the authors add XRD data for all samples to the manuscript.
Lines 93, 94. Why was the temperature range up to 1000 °С chosen? The pyrolysis of all components of the biomass ends at about 450 °C.
Lines 120. Which equation does “n” belong to? Please clarify.
Lines 233, 234. "The performance of the DTG curves shows similar behavior (Figure 2). Four main peaks can be seen in all of them ...". - The curves clearly show only two peaks. Authors should revise this paragraph.
Lines 244-247. I recommend rewriting this part to specify more precise temperature ranges.
Line 273. Table 3. Why is the interval from 150 to 1000 °С selected for lignin?
Lines 275-281. As the authors comment on the recommendations of the ICTAC kinetic committee? (H. Tanaka, ME Brown, J. Therm. Anal. Cal., 80 (2005) 795) ((Coats-Redfern, Freeman-Carroll and Horowitz-Metzger). Thermochim. Acta, 355 (2000) 125-143 and Thermochim. Acta, 520 (2011) 1-19)

Reviewer 2 Report

This manuscript reports on the validation of pseudo-component kinetic model on thermal gravimetric analysis (TGA-PKM) to determine the chemical compositions for various lignocellulosic materials. The authors carefully examined their method contrasting with chemical compositional analysis. Although the results of the analysis appear to be technically reasonable, I hope the authors should clarify the following points before considering of publication.

  1. What are the authors' views on the agreement between the chemical analysis results and the TGA-PKM method of lignocellulosic compositions (i.e., the errors shown in table 9)? Some fractions have very small errors within 1 wt%, however, some have quite as large as ~7 wt%. What level of accuracy are you aiming for?
  2. I think there is a trade-off between ease of operation and determination accuracy, but how efficient is the PKM method in determining composition compared to traditional chemical composition analysis in cost or analyzing speed?
  3. As thermal oxidation reactions are extremely complex reaction paths to occur the synergetic effect between the multiple components, the deconvolution of the DTG curve is only available as an approximation. In other words, I think the TG profiles are expected to vary greatly depending on conditions such as heating rates and sample particle size. However, only results under one type of condition are presented in this study. The validity of the PKM method for various measurement conditions should be verified.
